



# Brief Communication: Rapid high-resolution flood impact-based early warning is possible with RIM2D: a showcase for the 2023 pluvial flood in Braunschweig

Shahin Khosh Bin Ghomash[1], Heiko Apel[1], Kai Schröter[2], and Max Steinhausen[2]

[1]Section Hydrology, GFZ German Research Centre for Geosciences, Potsdam (Germany)
[2]Leichtweiß-Institute for Hydraulic Engineering and Water Resources, Division Hydrology and River Basin Management, Technische Universität Braunschweig, Beethovenstr. 51a, 38106 Braunschweig, Germany

**Correspondence:** Shahin Khosh Bin Ghomash, (shahin@gfz-potsdam.de)

**Abstract.** In recent years, urban areas have been increasingly affected by more frequent and severe pluvial floods, attributed to climate change and urbanization. This trend is expected to continue in the future, underscoring the critical importance of flood warning and disaster management. However, pluvial flood forecasts on a communal level do not exist in practice, mainly due to the high computational run-times of high-resolution flood simulation models. Here, we showcase the capability of the

hydrodynamic model RIM2D (Rapid Inundation Model 2D) to deliver highly detailed and localized insights into expected flood extent and impacts in very short computational processing times, enabling operational flood warnings. We demonstrate these capabilities using the case of the June 2023 torrential rain and resulting flood event in the city of Braunschweig, located in Lower Saxony, Germany. During this event, the city experienced intense rainfall of 60 liters per square meter within a short timeframe, resulting in widespread inundation, significant disruption to the residents' daily life, and substantial economic costs

to the city. This study serves as a clear indication that different dimensions of flood consequences can be simulated at very high resolutions in extremely short computational times and that models such as RIM2D, along with the necessary hardware for their operation, have reached a level of maturity suitable for integration into operational early warning systems and impact-based forecasting systems for such floods.

## 1   Introduction

Urban pluvial flooding presents a significant threat to many cities worldwide, and this threat is anticipated to increase due to the impacts of climate change and urbanization (Kundzewicz et al., 2014; Skougaard Kaspersen et al., 2017). During such events, precise flood information is crucial for emergency management decisions by local decision-makers, disaster managers, and potentially affected citizens (Tyler et al., 2021). Early forecasting systems play a crucial role in mitigating the loss of life and property during such occurrences, enabling the implementation of preventive measures ahead of time (Šakić Trogrlić et al.,

2022). Computational hydrodynamic models serve as essential tools in such systems, enabling the simulation and prediction of detailed flooding information under varying conditions.


Flood impact-based forecasting has recently gained attention within disaster risk research (Taylor et al., 2018; Zhang et al., 2019; Merz et al., 2020; Rözer et al., 2021; Apel et al., 2022). This approach aims to not only focus on metrics such as flood inundation area and water depths, but to expand forecasts to encompass additional and arguably more comprehensible

dimensions of impacts, such as flow velocity, the number and locations of affected individuals and buildings, damage to structures and infrastructure, or disruptions to services. It is suggested that when such distinct and spatially detailed information in addition to behavioral instructions are disseminated to the public, individuals are more likely to heed warnings and respond in a more effective manner before or during a flood event (Kreibich et al., 2021). Typically, determining such impacts necessitates the use of distributed water depth data from the flooding, alongside potential additional factors such as velocity, duration,

precautionary measures, and warning times (Merz et al., 2010; Jongman et al., 2012). Physically-based distributed models allow for the simulation of such parameters, thereby enabling the calculation of flood impacts at the desired level of detail.

Physically-based hydrodynamic models, typically based on the two-dimensional Shallow Water Equations, have a long-standing history in flood modeling, with various applications showcased under different scenarios e.g. (Pasculli et al., 2021; Khosh Bin Ghomash et al., 2022; Apel et al., 2024). They hold promise in improving the accuracy and efficiency of early

forecasting systems (Apel et al., 2022; Cea and Costabile, 2022; Costabile et al., 2023). However, the practical integration of such models into early warning systems has, until recently, been significantly limited due to challenges associated with computational capacities, data assimilation, and real-time decision-making (De Almeida and Bates, 2013; Hill et al., 2023). Yet, this is evolving due to advancements in high-performance parallel-computing (HPC) systems in recent years. These systems have facilitated physically-based flood modeling with an unprecedented level of detail, all while requiring reasonably short com-

putation times (Caviedes-Voullième et al., 2023). RIM2D ("Rapid Inundation Model 2D") is a GPU-accelerated 2D hydraulic simulation model, written in CUDA FORTRAN, capable of extensive parallelization on HPC systems equipped with CUDA-enabled NVIDIA Graphics Processing Units (GPUs). It solves a simplified version of the Shallow Water Equations (Bates et al., 2010) and has been effectively employed for simulating floods across various scenarios (Apel et al., 2016, 2022, 2024; Khosh Bin Ghomash et al., 2024).

The question that naturally follows is: Can high-performance computing-enabled shallow water solvers such as RIM2D attain adequate accuracy and lead time to enhance early flood warning systems for more effective management of events? In this study, we present the workflow for simulating various dimensions of flood impacts using RIM2D, and seek to answer if computational times are sufficiently short for use of such models in impact-based operational early flood warnings. We showcase this for the 2023 pluvial flooding in Braunschweig, a city in Lower Saxony, Germany. In June 2023, Braunschweig

experienced a notable instance of pluvial flooding, during which 60 liters per square meter of rainfall occurred within a brief timeframe. This amount surpassed the capacity of the city's sewer system, resulting in widespread inundation across the urban area. The city area comprises densely populated urban zones, encompassing residential neighborhoods, industrial sectors, and critical infrastructure, which highly heightens the susceptibility to such urban pluvial flooding within the city. In this study, we showcase RIM2D's capability to simulate the 2023 flood event and its subsequent impacts on residential buildings and

impacted people. Model validation is performed With the use of volunteered geographic information (VGI) (Drews et al., 2023), and the resulting model runtimes are evaluated for their suitability of use in an operational pluvial flood forecasting



system. furthermore flood impacts in terms of monetary damage and affected people were derived in order to illustrate the benefits of spatially explicit flood impact forecasts.

## 2 Methods

### 2.1 Hydrodynamic model: RIM2D

RIM2D is a 2D raster-based hydrodynamic model developed by the Section Hydrology at the German Research Centre for Geoscience (GFZ) in Potsdam, Germany. RIM2D is designed to solve the local inertia approximation of the Shallow Water equations (Bates et al., 2010), demonstrated to be effective for flood inundation modeling (Falter et al., 2014; Neal et al., 2011; Apel et al., 2022, 2024). The local inertia formulation offers a more accurate representation of the issue compared to simpler versions of the Shallow Water equations, such as the diffusive wave model (De Almeida and Bates, 2013; Caviedes-Voullième et al., 2020). It introduces an additional term to account for the rate of change of local fluid momentum, influencing how the fluid's momentum evolves from one-time step to the next. In practical terms, this means that the fluid's momentum at a given time step informs its behavior in the subsequent step, necessitating an acceleration of flow from its previous state. Thus, in the realm of describing shallow water flows, the local inertial formulation bridges the gap between the diffusive wave approximation and the more comprehensive full-dynamic equations.

While the original explicit numerical solution proposed by Bates et al. (2010) can be prone to instabilities under conditions of near-critical to super-critical flow and for small grid cell sizes (De Almeida and Bates, 2013), RIM2D has integrated the numerical diffusion method put forth by de Almeida et al. (2012) to address these challenges.

RIM2D is implemented in Fortran and has been adapted for use with GPUs through CUDA Fortran libraries. It should be noted that, at present, RIM2D supports computations on a single GPU. However, ongoing efforts are focused on incorporating multi-GPU computing capabilities into RIM2D soon.

### 2.2 Data and model setup

The digital elevation model (DEM) utilized for the model setup was the DGM1 product, provided by the Braunschweig city administration and generated by the Federal Agency for Cartography and Geodesy (BKG) of Germany. This DEM, with a cell resolution of 1 meter, was produced using lidar mapping data from 2016 covering the city area (192.1 $km^2$). To explore the efficiency of RIM2D under different resolutions and to lessen the computational load of the simulations, the DGM1 dataset was resampled to resolutions of dx = 2, 5, and 10 meters using the averaging method. This resulted in raster comprising approximately 75 million cells for a resolution of dx = 2 meters, 12 million cells for dx = 5 meters, and 3 million cells for dx = 10 meters within the study domain. Additionally, building outlines sourced from OpenStreetMap were used to remove buildings footprints from all three DEMs. As a result, the surfaces of the buildings served as closed reflective boundaries within the simulations.





Manning roughness values were assigned to the study domain based on the 2020 Germany land cover classification, derived from Sentinel-2 data (Riembauer et al., 2021). This classification is based on atmospherically corrected Sentinel-2 satellite imagery processed with the MAJA algorithm, with data provided by the EOC Geoservice of the German Aerospace Centre (DLR).

Additionally, training data from other reference sources such as OpenStreetMap, as well as the Sentinel-2 scenes themselves, were used in the classification process. This specific land cover dataset was selected for the study due to its relatively detailed land classification and fine grid resolution of 10 meters. Manning values of 0.04, 0.03, 0.1, 0.025, 0.035, 0.035 [$m^{-1/3}s$] were assigned to the vegetation, water bodies, forest, built-up, bare soil and agriculture land categories respectively.

For the model's precipitation boundary for the 2023 flood event, the rainfall radar data "RADOLAN" provided by the German

Weather Service (DWD) was used. RADOLAN, an operational open-source product of the DWD, is accessible at the Climate Data Center (CDC) of the DWD. This data, available at hourly intervals, covers all of Germany at a spatial resolution of 1,000 × 1,000 meters. 48 sequences of hourly RADOLAN images, corresponding to the days June 22nd and 23rd, 2023, covering the extent of our study domain, was used as boundary conditions for the model. It has to be noted that the hourly resolution can underestimate sub-hourly precipitation intensities. DWD provides also reprocessed 5 Minute resolution RADOALN products,

but these are available with a latency of about one year, and are thus not available for this study.

Infiltration rates for the domain surface were determined through the combination of two distinct datasets. Initially, the soil type within the study area was classified using the BK50 soil map (Evertsbusch et al., 2020) (with a resolution of dx = 50 meters) provided by the State Agency for Mining, Energy, and Geology of Lower Saxony, Germany. Saturated hydraulic conductivity values suitable for each soil category in the BK50 raster were assigned following a comprehensive literature review.

Subsequently, the Imperviousness Density percentage raster, sourced from the Earth Observation component of the European Union's space program (Copernicus) at a resolution of dx = 10 meters, was utilized to compute the surface perviousness across the study domain. The final infiltration rates integrated into the model were derived by multiplying the saturated hydraulic conductivity of each cell based on the BK50 product with the surface pervious percentage of that cell. RIM2D also enables the consideration of the urban sewer drainage system, utilizing a capacity based approach analogously to the infiltration. This

approach has been proven to provides reliable simulation of drainage volumes and thus of the impact of the sewer drainage on the surface inundation (Apel et al., 2024). However in this study the sewer capacity was not considered in order to make the simulations comparable with the available city flood hazard maps.

## 2.3  Impact Calculation

We use the loss estimation model developed by Rözer et al. (2019) derived from empirical surveys conducted via computer-

aided telephone interviews in five German cities following pluvial flood events from 2005 to 2014. This model expresses loss as relative loss (rloss) between 0 meaning no damage to the building and 1 complete destruction of the building. Relative loss is then multiplied by the building's asset value to calculate monetary losses. The model is only trained to predict losses for residential buildings. To predict a building's relative loss due to pluvial flooding, the model employs Bayesian zero-inflated beta regression (Ospina and Ferrari, 2010), considering factors such as water depth, duration, contamination, presence of





a basement, and household size as influences of damage. The model is implemented in R using the Rstan package (Stan Development Team, 2024) with MCMC sampling.

For the case of Braunschweig, the damage calculation is based on maximum water depth values and the water timing in the domain which is the duration for which each cell in the area flooded with a minimum water depth of 1 cm. To estimate losses in Braunschweig, OpenStreetMap data for building footprints is used. Several filters were applied to the raw OSM dataset to

ensure that only residential buildings are included in the analysis. Buildings that meet the following criteria were assumed to be residential and therefore included in the loss estimation:

- Buildings without a name tag, as the assumption is that only prominent non-residential buildings are commonly tagged with names by OSM contributors.

- Buildings without a type tag, since most buildings, 80%, do not have a specific type assigned.

- Buildings with the following type tags: "residential", "detached", "apartments", "house", "villa", "terrace", "bungalow", "semidetached-house", "allotment-house", "apartments;yes", "winter-garden", or "prefabricated".

- Buildings with a footprint size smaller than 3000 $m^2$, as it is assumed that buildings larger than this are likely industrial or commercial, not residential in the area of interest.

Monetary asset values for residential buildings are based on HANZE data (Paprotny et al., 2018). This data was derived by

distributing the national stock of dwellings to regions (NUTS-3 level) based on regional gross domestic product (GDP). Further, the estimated regional stock was disaggregated proportionally to the population per grid cell (Steinhausen et al., 2022), and then adjusted based on the OSM footprint areas. These asset values are adjusted to 2023 prices and should be interpreted as reconstruction cost.

In addition to assessing damages, for illustrative purposes, we also compute the number of individuals affected during the

flood event. Those considered affected include all individuals residing within areas impacted by flooding ("people who get wet feet"). The potential impact is determined by the population density [person/m2] corresponding to each cell within a raster that delineates the modeling area. Vulnerability is defined by the level of impact, which is determined by the maximum water depth [m]. To mitigate uncertainties associated with very low water depths, we establish a threshold of 10 cm for this calculation, meaning that if the water depth falls below 0.1 m, no individuals within the cell are considered affected; On the other hand, if

the water depth exceeds 0.1 m, all individuals within the cell are assumed affected. For this calculation, we make use of the latest WorldPop population density map (Tatem, 2017) for Germany for the year 2020, with a resolution of dx = 100 meters.

## 2.4 Model Validation

Field observations relevant to urban pluvial floods, such as water level measurements, are rarely accessible or, at most, extremely sparse (Francipane et al., 2021; Drews et al., 2023). This scarcity stems from the brief duration and localized nature of

intense rainfall, which reduce the chances of mapping these events by operational remote sensing platforms. And even if remote sensing platforms observe the flood event, problems mapping the flood arise from cloud cover (optical sensors) or reflections





from buildings (radar sensors). The infrequent occurrence of heavy precipitation events is an additional obstacle in obtaining pluvial flood maps for model validation. Consequently, there is typically a very limited quantity and quality of observational data from real-world pluvial flood events available for validating advanced urban flood models. In fact, in numerous locations

that are theoretically at high risk of pluvial flooding, there are no historical records whatsoever. While the hydrodynamic model RIM2D has been validated across various flooding scenarios, including both pluvial (Apel et al., 2016, 2024) and fluvial cases (Apel et al., 2022; Khosh Bin Ghomash et al., 2024), this study intentionally conducts simulations in a blind manner, without parameter calibration such as roughness coefficients. This approach is chosen primarily because of the limited availability of data from the city and the specific nature of these floods, but also to assess the feasibility of using solvers like RIM2D for early

warning systems, recognizing that comprehensive calibration may not always be feasible for every city covered by the early warning system. Therefore, relying on available spatial data and standard parameterizations is often the only practical option. While calibration would be ideal, the typical absence of calibration data (such as flood mapping) and the occasional need for rapid model deployment in operational scenarios mean that using an uncalibrated model is common practice. Thus, this study presents uncalibrated simulation results without delving into extensive model calibration efforts.

As there were no systematic observations from the 2023 flood event in Braunschweig, the sole data that could be used for model validation were VGI data, such as pictures and videos captured by the residents of Braunschweig during the flood event. Recordings of the flood in 164 different locations within the city were provided by the city administration. Studies such as Assumpção et al. (2018) and See (2019) indicate that although the quantity of data gathered in most VGI case studies may not be extensive, however, it still can offer an effective means of validating pluvial flood models at urban scales. In this paper, we

utilize these recordings as references, employing them to qualitatively validate the flood extent and inundation depths of the RIM2D model for the simulation of the 2023 flood. To further validate our model, we utilize the city's official flood hazard maps, which are provided by the city administration and cover events with return periods of 30, 100, and 1000 years. These maps, derived from hydrodynamic simulations, have undergone rigorous verification and validation by experts knowledgeable about the city and its historical flood events. A comparison of our model results with these city maps is presented in Appendix A.

# 3  Results and discussion

## 3.1  Flood Impacts

Next, we turn to the simulation of the 2023 flood event. Figure 1 illustrates the maximum water depth (a), maximum water velocities (b), the number of affected people in the domain (c), and the resulting flood damages to residential buildings (d). These results are based on the dx = 5 m configuration. This figure showcases RIM2d's ability to generate impact-based con-

sequences of the 2023 flood in a spatially distributed manner. Concerning each map, from Figure 1a it can be seen that high water depths (over 50 cm) are noticeable both in the city center, characterized by densely packed urban areas and a significant population concentration, as well as in the surrounding areas. With regards to flow velocity (Figure 1b), relatively low velocities are observed across the entire domain, something which is common in the case of pluvial floods in rather flat terrains such as Braunschweig. A higher concentration of affected individuals (light green) is observed in the central part of the city (Figure 1c)





in comparison to the outskirts of Braunschweig which is mainly due to the high population density in the city center. Similarly, residential building losses predominantly occur in the city center (seen in Figure 1d), particularly in highly urbanized areas. Maps, as presented here, enhance flood forecast information to include additional and more comprehensible dimensions of impacts.

Regarding the June 2023 flood event in the city, no systematically measured observational data were available from the
flood. To evaluate the accuracy of the RIM2D simulation, we conduct a qualitative comparison of RIM2D results with social media posts recorded by the city residents during the event, provided to us by the city administration. We mainly compare flood extent, but also where possible, water depth. Figure 2 presents the 164 locations were photo and videos were available and two examples of this comparison. RIM2D results are visualized in GoogleEarth to facilitate the assessment. The lower comparison figure showcases flooding in Willy Brandt Platz in front of the city's main train station (Hauptbahnhof), while the upper figure
depicts flooding on the Bohlweg street within the city center. Some further examples of visualized comparisons are presented in Figure B1 in Appendix B. Furthermore, 30 out of the 164 locations, which were the most understandable and measurable in terms of flood water depth, were identified. Reference objects such as street poles, house corners or cub stones seen in the photos and videos were then manually measured with a DGPS (Differential Global Positioning System) in the city, and water depths were estimated accordingly. Figure B2 in Appendix B presents a comparison between these estimated water depths and
the simulated depths using RIM2D. Although it is observed that RIM2D occasionally overestimates the water depth, this can be attributed to several factors, one of which is the exclusion of the sewer system in the model. Incorporating the sewer system could potentially improve the accuracy of the results. Overall, in almost all compared instances, RIM2D results are closely aligned, both in terms of flooded areas and water depth (where comparison was possible), with the observations depicted in the photos/videos. This indicates that it was possible to predict the potential flooded areas of the event using RIM2D, highlighting
the effectiveness of RIM2D as a valuable early warning tool.

## 3.2 Computational performance and runtime

We continue by examining computational performance, as runtime and computational resources are critical factors in determining the applicability of this technology for flood forecasting and early warning systems. Figure 3 illustrates the absolute simulation runtimes (a) and the ratio of the simulated period to the simulation runtime (b) for simulations at dx = 10, 5, and 2
meters. The red line represents the values for the 2023 flood event (48 hours simulation period), while the blue line represents the values for the 100-year event (2 hours simulation period). As the simulation resolution increases, the runtime also rises significantly, with the dx = 2 m simulation requiring more than 33 times the computational time compared to dx = 10 m.

Nevertheless, even at the finest resolutions, RIM2D proves to be suitable for integration into early warning systems. For the 2023 flood event at dx = 10 m, RIM2D operates more than 211 times faster than the real-time event. Even at the extremely
fine dx = 2 m resolution, comprising 75 million cells, simulations can still be conducted more than 6.5 times faster than the actual event. This high level of efficiency makes RIM2D ideal for improving current operational flood forecast systems while maintaining impressive forecast lead times. Consequently, this capability enables detailed flood impact forecasting and effective responses. It should also be noted that estimating other impact dimensions such as losses and determining the affected





population require computation times within seconds, alleviating any additional computational burden on the calculations. In

an actual operational forecasting and warning system, updating the simulations would involve simply adjusting the rainfall boundary to the latest rainfall images to generate a forecast. Subsequently, impact calculation results would be derived from this updated data.

It's important to note that no specific optimization of the models was conducted to maximize performance for this particular scenario. Such optimization would be necessary for operational purposes, potentially enhancing performance to some

extent. Furthermore, ongoing advancements in solver implementations, such as introducing a multi-GPU solver for RIM2D, are expected to further enhance computational efficiency.

## 3.3   Uncertainties

Like all model simulations, there can be inherent uncertainties in the results. The uncalibrated hydrodynamic model presented here, will of course add to the existing uncertainties in the meteorological forecasts. However, hydrodynamic modeling is

known to introduce the least amount of uncertainty compared to other sources (Apel et al., 2009; de MOEL and Aerts, 2011). This uncertainty could be further minimized by avoiding ad hoc setups and uncalibrated runs, as seen in this feasibility study, and instead opting for a more detailed implementation that includes calibration or validation against historical flood events or other sources such as volunteered geographic information (VGI) (Drews et al., 2023). Achieving such an improved model implementation would be feasible with the knowledge and cooperation of the city's responsible authorities. When implementing

RIM2D in an operational context, it is recommended to include uncertainty maps generated from an ensemble of meteorological forecasts. This approach helps to visualize the inherent uncertainties. Leveraging the computational efficiency of RIM2D, these meteorological forecast ensembles can then be transformed into inundation maps with uncertainty information associated to the flood forecast (Alfonso et al., 2016; Zarzar et al., 2018). These maps can then illustrate the potential outcomes of uncertainties in the meteorological forecasts, thereby representing the uncertainty within the inundation forecast.

## 4   Conclusions

This study utilizes the recent 2023 pluvial flood event in Braunschweig, Germany, to showcase the capabilities of the hydrodynamic model RIM2D for flood impact forecasting and the potential advantages of such forecasts. We demonstrate that RIM2D provides credible simulations of inundation areas, depths, and flow velocities within efficient runtimes, enabling timely forecasts and early warnings. Additionally, we can derive supplementary impact indicators that pinpoint hazardous areas, such as

those with expected damages or affected populations. This detailed, site-specific data on expected impacts is immensely useful for precise and focused flood disaster management. It provides the potential for more informative public alerts than traditional flood depth maps, potentially resulting in significant improvements to current disaster response and warning systems.

The use of graphical processor units (GPUs) optimized for massive parallel computing enables RIM2D simulations to achieve runtimes compatible with integration into operational flood early warning systems. We show that for this particular

event, it is currently possible to generate flood forecasts 211 times faster-than-real-time at 10 m resolution, 50 times faster-





than-real-time at 5 m resolution and 6.5 times faster-than-real-time at 2 m resolution with a single scientific grade GPU. In particular, the dx = 5 and 10 meter setups can complete their runs in 13 and 58 minutes, respectively, means within the span of an hour. Using 4 GPUs the simulation runtimes can be further reduced to 7, 29 minutes for dx = 10 and 5 m respectively.

This holds particular significance, especially in the case of floods, where the type and timing of flood warnings are determining factors in the extent of the damages and casualties of such events. Historically, numerous regions depend on early warning systems which typically lack spatial detail and are issued on a district or municipal level and primarily only utilize rainfall data for communication. Nevertheless, instances like the flood in Braunschweig underscore the possible drawbacks of such systems, particularly in situations where details about water levels and affected areas are crucial for efficient response and decision-making. The model employed in this study demonstrates the capability to simulate water levels and flooded areas

with a high degree of detail. With its straightforward model setup and the widespread availability of the required input data in many countries, RIM2D shows promise for widespread adoption and utilization. Additionally, the hardware required for running these simulations is cost-effective, especially in comparison to large-scale computational clusters. This affordability facilitates the implementation of the model at flood forecast centers without requiring substantial investments in complex IT infrastructure. Moreover, as most of the existing cloud-based HPC systems like Google, Amazon and Deutsche Telekom host

GPU computing services, the technical infrastructure can also be outsourced at comparatively low costs.

Considering the validity of the simulations, the ease of implementing hydraulic forecast models, and the speed of the simulations, we propose extending current forecast practices by incorporating impact forecast models, such as the one presented here. This expansion could enhance the effectiveness of flood warnings and management. The scientific foundation, methodologies, and models for these impact forecasts are well-developed and ready for implementation within operational forecast systems.

In conclusion, the primary result of this study is evidence that this technology has reached a level of maturity suitable for integration into early warning systems. This sufficient lead times for many emergency measures and delivers significantly more detailed and actionable outcomes compared to conventional flood early warning systems. The detailed and time-sensitive depth and velocity data and further categories of impact offer a clearer anticipation of flood severity, enabling further analysis to derive impact estimation. Moreover, this information is far more accessible to the general public, managers, and emergency

responders compared to alerts based solely on, for instance, rainfall data (Thieken et al., 2023).

*Code availability.* RIM2D is open-source for scientific use under the EUPL1.2 license. Access is granted upon request. The simulations were performed with version 0.2.

*Data availability.* The OSM building shape files used in this research can be freely obtained from https://download.geofabrik.de/europe/germany.html. The land cover raster, which was used to assign roughness values to the simulation domain, is openly accessible at https://www.mundialis.de/en/germany-

2020-land-cover-based-on-sentinel-2-data/.



*Author contributions.* SKBG: Conceptualization, Methodology, Investigation, Simulation, Software, Analysis, Visualization, Writing; HA: RIM2D Software, Conceptualization, Review, Writing; KSL: Conceptualization, Review, Writing; MS: Conceptualization, Simulation, Software, Review

*Competing interests.* At least one of the (co-)authors is a member of the editorial board of Natural Hazards and Earth System Sciences

*Acknowledgements.* This research was performed within the frame of the DIRECTED project (https://directedproject.eu/). Funding of the DIRECTED project within the European Union's Horizon Europe – the Framework Programme for Research and Innovation (grant agreement No. 101073978, HORIZON-CL3-2021-DRS-01) is gratefully acknowledged. This work utilized high-performance computing resources made possible by funding from the Ministry of Science, Research and Culture of the State of Brandenburg (MWFK) and are operated by the IT Services and Operations unit of the Helmholtz Centre Potsdam. We thank the contributors of VGI data in Braunschweig and Lennard
Marks for his contributions to the validation dataset.

**Appendix A: Supplementary Material: Model comparison with official city flood hazard maps**

Initially, the city's official flood hazard maps (provided by the city administration) were utilized, comprising max water depth maps for events with return periods of 30, 100, and 1000 years. These maps had been produced with the HYDRO-AS-2D (Version 5.2.4) hydrodynamic model with the basis of the DGM1 elevation product (also used in this study with RIM2D)
under spatially-temporally uniform standard design rainfall as forcing. The rainfall depths associated with these events are 41.4 mm, 50.6 mm, and 75 mm, respectively, each lasting for 1 hour. The flood maps were constructed through simulations involving one hour of rain followed by one hour of runoff. We apply identical boundary conditions to the RIM2D model for Braunschweig. To facilitate the comparison of the inundated areas of the RIM2D simulations with the official city maps, an extensive set of metrics is employed. These metrics are computed by evaluating the maximum inundation maps of the RIM2D
simulations against the official city flood maps. Initially, cells are categorized based on Table A1. For this classification, RIM2D results are considered as simulated and the city flood maps are considered as observed. A confusion map is generated from each comparison, providing the total counts for the indices presented in Table A1. These counts are then utilized to compute the domain-wide inundation metrics as illustrated in Table A2. These metrics are based on works by Wing et al. (2017) and Bernini and Franchini (2013).
Figure A1 illustrates the Critical Success Index, Hit Rate, False Alarm, Error Bias, and Bias Percentage Indicator for comparing flooded areas between the outputs generated by RIM2D and the official flood hazard maps of Braunschweig. The results are presented for the 30, 100, and 1000 yearly events, each at resolutions of dx = 2, 5, and 10 meters. In general, the results show a strong alignment between the RIM2D outputs and the official flood hazard maps of Braunschweig, with high scores across all indicators when considering flooded areas. It is important to note that the city's official flood hazard maps are also
the product of hydrodynamic simulations, however, they have been robustly verified and validated by experts familiar with the





|  |  | Simulated | |
|---|---|---|---|
|  |  | Wet | Dry |
| Observed | Wet | True Positive (TP) | False Negative (FN) |
|  | Dry | False Positive (FP) | True Negative (TN) |

**Table A1.** Inundation confusion matrix. Each cell in the domain for a given simulation is compared to the corresponding cell in the observed grid and classified according to this table.

| Metric | Equation | Poor | Perfect | Description |
|---|---|---|---|---|
| Critical Success Index | $\dfrac{TP}{TP+FP+FN}$ | 0 | 1 | ratio of accurate wet cells to total wet cells and missed wet cells |
| Hit Rate | $\dfrac{TP}{TP+FN}$ | 0 | 1 | portion of observed wet cells reproduced by the model |
| False Alarms | $\dfrac{FP}{TP+FP}$ | 1 | 0 | portion of modelled wet cells which are erroneous |
| Error Bias | $\dfrac{FP}{FN}$ | 0 or inf | 1 | ratio of over-predictions to under-predictions |
| Bias Percentage Indicator | $100\left(\dfrac{TP+FP}{TP+FN}-1\right)$ | -100 or 100 | 0 | relative percentage error of the final extent of the flooded area |

**Table A2.** Flood inundation performance metrics

city. In regard to the simulated event (30, 100, or 1000 years), there is minimal impact observed on the scores. The resolution of the simulations however has a slight effect, with higher resolutions yielding slightly higher scores compared to coarser setups. However, even at the coarse resolution of dx = 10 meters, high scores are achieved, indicating the suitability of this model resolution for pluvial flood modeling in the city. Since these indicators solely assess the alignment of flooded areas, to assess

the differences in simulated water depth values, we calculate the Root Mean Squared Error (RMSE) in the inundated areas between the water depths of the official city maps and the RIM2D simulation results at dx = 2 meters. This yields an RMSE of 0.075 [m], indicating a strong match. It is important to mention that the existing RIM2D model is not calibrated, and these results could be even further enhanced through additional calibration efforts. It also has to be noted that the largest differences compared to the reference simulations occur in a few locations at the outskirts of the city, where culverts are present. As these

are not considered in RIM2D, differences in the simulated inundations depth greater than the RMSE difference are observed. These are, however, very localized.

**Appendix B:  Supplementary Material: Examples of model comparison with VGI data**



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

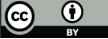

**Figure 1.** (a) Simulated max water depths [m], (b) max velocities [m/s], (c) affected population [number of Persons] and (d) loss incurred to residential buildings [Euros] for the 2023 flood event at dx = 5 m.

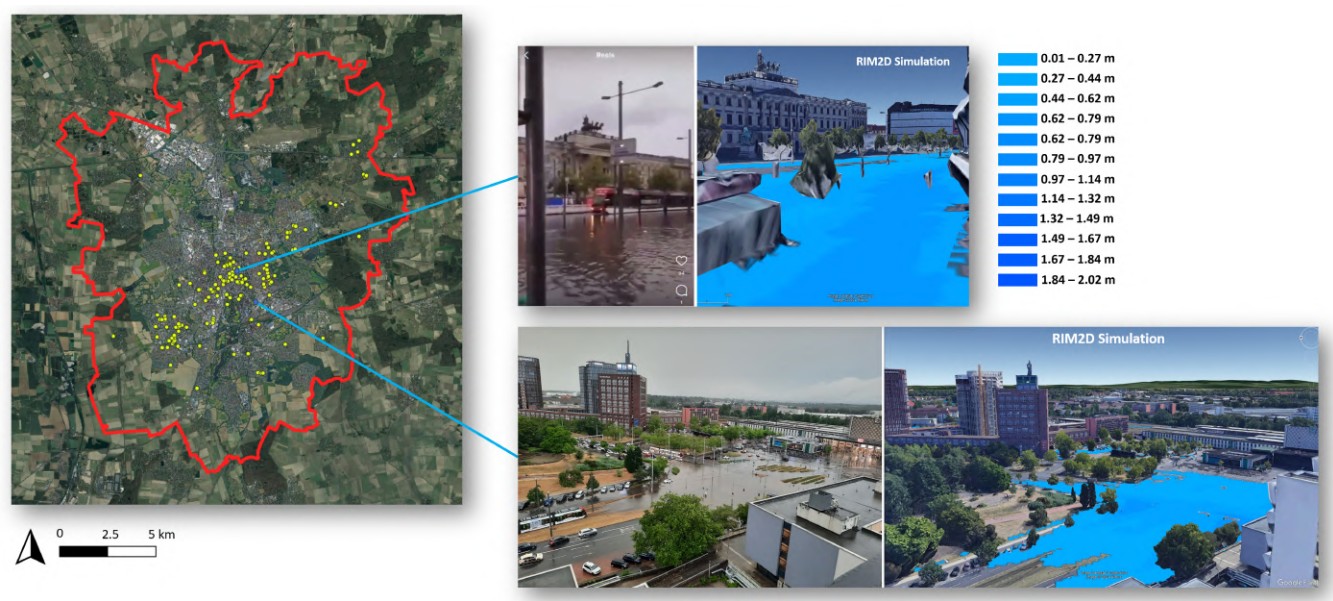

**Figure 2.** Red line represents Braunschweig city administrative boundary and yellow dots show the 164 locations where photos/videos from the 2023 flood were available. Two examples of comparisons between the dx = 5 m RIM2D simulation results to social media posts are presented. Visualization of the RIM2D results and the background imagery are performed with and provided by Google Earth. Satellite imagery: © Google Earth 2024


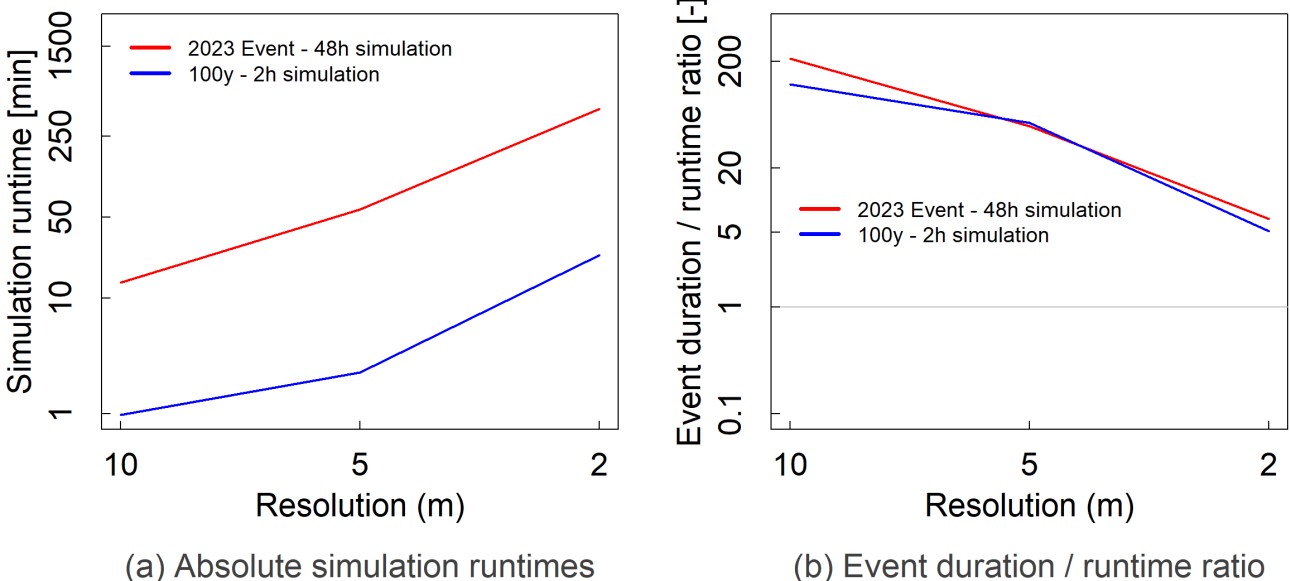

(a) Absolute simulation runtimes

(b) Event duration / runtime ratio

**Figure 3.** The absolute simulation run-times (a) and the ratio of the simulation duration to the simulation runtimes (b) for the dx = 10, 5, 2 m simulations. Red line indicates the values for the 2023 flood event (48 hours of simulation) and blue line represents the values for the 100-year event (2 hours of simulation).

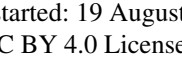
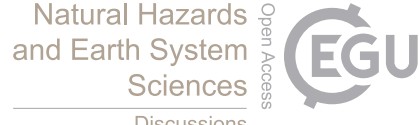


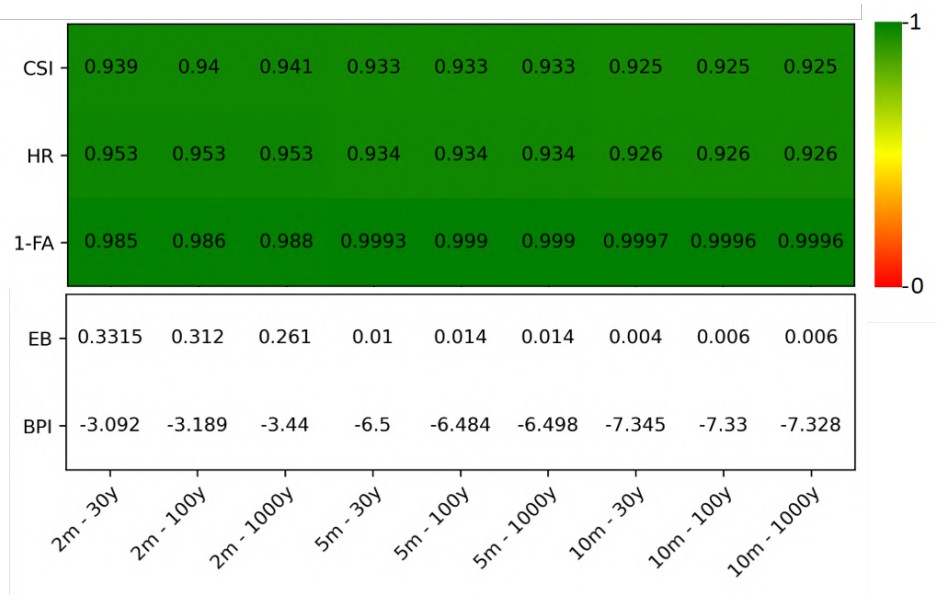

**Figure A1.** Comparison of RIM2D flooded areas to the official city flood maps for resolutions dx = 2, 5 and 10 meters under 30, 100 and 1000 yearly event scenarios. The indices Critical Success Index, Hit Rate, False Alarm,Error Bias and Bias Percentage Indicator are presented for the comparison
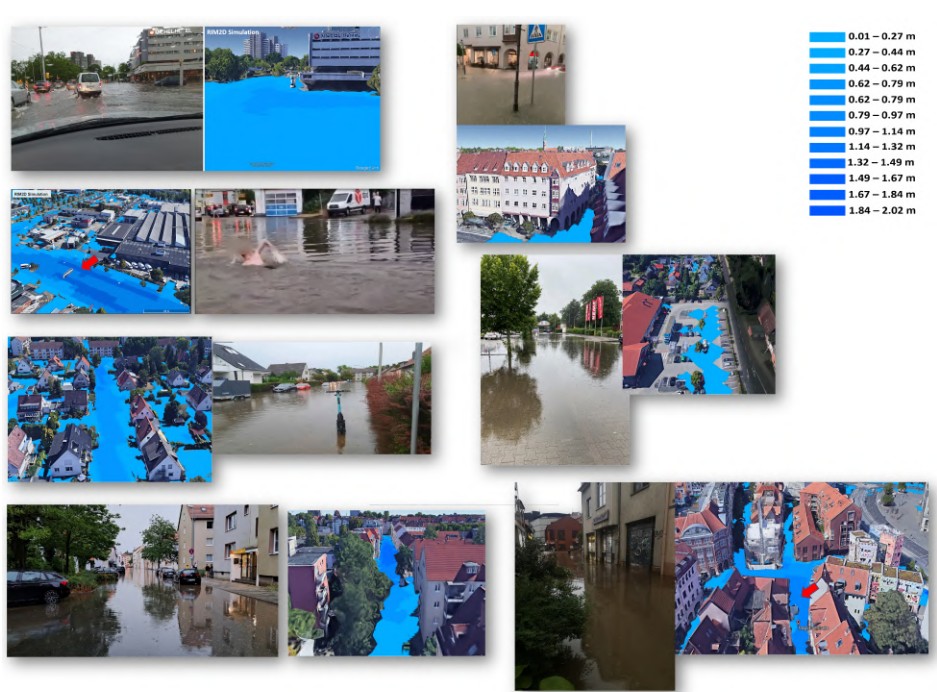

**Figure B1.** Examples of comparisons between the dx = 5 m RIM2D simulation results to social media posts. Visualization of the RIM2D results and the background imagery are performed with and provided by Google Earth. Satellite imagery: © Google Earth 2024



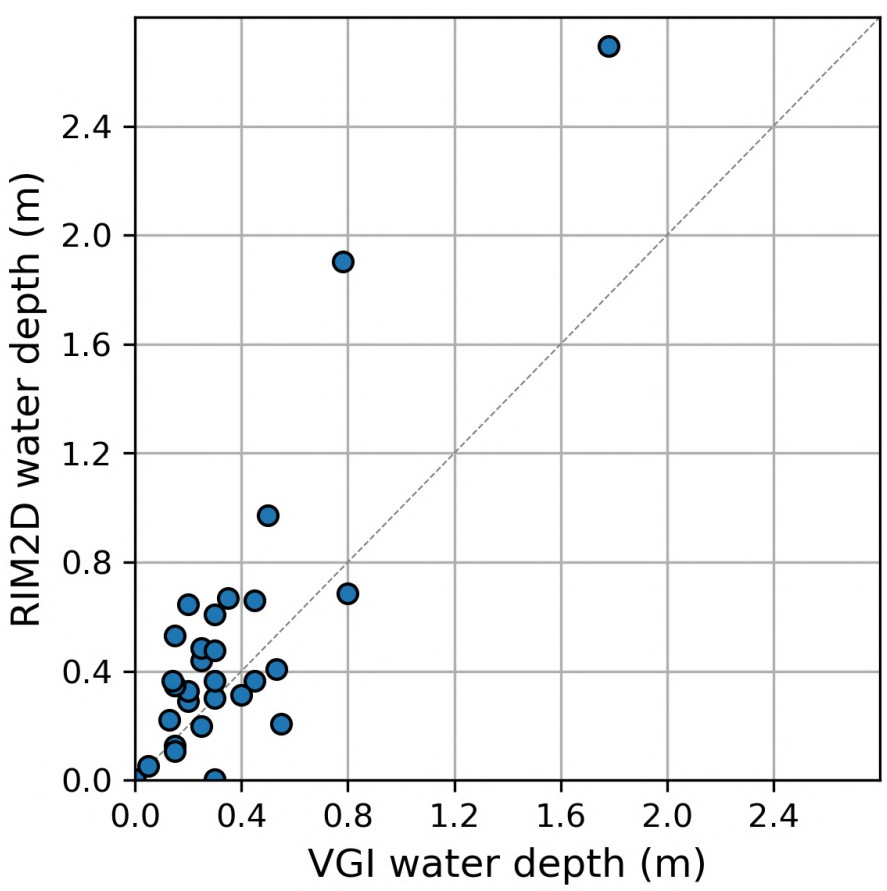

**Figure B2.** Validation of the dx = 5m RIM2D simulation water depths at 30 VGI validation locations in Braunschweig using measure water marks