# Peer review of "Rapid high-resolution flood impact-based early warning is possible with RIM2D: a showcase for the 2023 pluvial flood in Braunschweig"

_Natural Hazards and Earth System Sciences, 2024_

## Author Response (AR1)

**REPLY on RC1**

**Dear RC1,**

Thank you for reviewing our manuscript and providing valuable feedback. We appreciate your positive comments and have addressed your remarks inline below.

This study shows a newly developed/combination technique that has the potential to enhance flood early warning systems based on the level of impacts. The topic is important, the writing is clear, the logic is sound, and the results appear to be significant. Overall, I suggest only minor revisions are needed before it can be considered for publication in NHESS as a Brief Communication paper. Specifically, I have a few comments:

1- While I understand the challenges of quantitatively validating the modeling results due to the lack of systematic data, such validation is crucial before asserting that the predictions of this technique are evident, as the authors claim in the conclusion. Could the authors explore indirect approaches to quantify the results? For instance, calculating the ratio of coherent versus incoherent grids based on the predicted and observed inundation areas (using only the grids with observations) could be akin to a confusion matrix analysis.

**REPLY:** Thank you for the comment. As described in the manuscript, for the 2023 flood event in addition to the qualitative comparison of the results to social media posts, we did conduct a qualitative validation by manually estimating water depths at 30 of the 164 locations where social media posts of the flood were available. This involved visiting the city and measuring reference objects like street poles, house corners, and curbstones visible in the photos and videos, focusing on the locations that provided the clearest and most measurable information. The results of this validation are shown in Figure B2. While calculating the ratio of coherent versus incoherent grids and comparing flood extent for the 2023 event, as suggested, would be a valuable validation approach, the complexities involved — such as varying image/video quality or different viewing angles in each of the image/video files — would make this process highly challenging. Moreover, this process would require specialized expertise in photogrammetry, which is beyond our current capabilities. We have, of course, compared the RIM2D results to the city's official flood hazard maps using advanced indicators that assess coherent and incoherent grids based on the predicted and observed inundation areas. The results of this comparison are presented in Figure A1.

Additionally, we have now added some sentences to the manuscript emphasizing your suggestion and highlighting the significance of such comparisons in instances where data is available for validating flood simulations in future studies.

**2- As the authors suggest, providing an uncertainty map for this tool is both important and useful. Why was this map not included in the exemplary case presented in the study?**

**REPLY:** As mentioned in the manuscript, including uncertainty maps generated from an ensemble of meteorological forecasts would indeed enhance the reliability of flood predictions in flood risk assessments. However, their omission in this case study was intentional, as the focus was on evaluating the feasibility of the RIM2D model for rapid forecasting rather than performing a full-scale operational implementation. The aim was to demonstrate the model's

runtime efficiency and impact estimation capabilities. Additionally, since the simulation targeted the specific 2023 storm event using observed rainfall data (or storm-specific flood maps, comparison results presented in Figure A1), we did not prioritize creating uncertainty maps based on forecast ensembles. That said, given the computational efficiency of RIM2D, incorporating uncertainty maps in future studies would be highly beneficial for presenting a more comprehensive view of uncertainties.

3- The manuscript currently contains over 20 references, which exceeds the limit for a Brief Communication in NHESS. Additionally, there are a few typographical errors that need correction.

**REPLY:** Thank you for bringing this to our attention. We have now reduced the number of references in the manuscript by selectively removing some while retaining those that are essential for supporting our findings and methodology. We will, of course, make further reductions if the current number is not acceptable to the editor. Additionally, we have conducted a thorough proofreading of the manuscript to correct any typographical errors.

**REPLY on RC2**

Dear RC2,

Thank you for reviewing our manuscript and for your positive feedback. We have provided our responses inline below.

The paper analyzes the capabilities of the RIM2D hydrodynamic model for flood impact forecasting, using the 2023 pluvial flood in Braunschweig, Germany, as a case study. The authors demonstrate that RIM2D produces accurate simulations of inundation areas, depths, and flow velocities with computational efficiency, supporting timely forecasts and early warnings. They evaluated computation times for a 48-hour flood episode and 2-hour design floods, showing that such analyses can be effectively integrated into real-time flood early warning systems. The study's main scientific contribution is the evidence that this technology has reached sufficient maturity for operational use in early warning systems. While the results are specific to the studied location, the methodology could be adapted and applied to other areas to validate these findings further.

In my opinion, the research is relevant and deserves publication in Natural Hazards and Earth System Science. The topic is interesting and fits in the scope of the journal, the objectives are clearly defined, previous work is properly presented and acknowledged, the methodology is clearly explained, the results are adequately described and discussed, and the conclusions are useful for many scientists interested in the topic. Overall, the paper is clearly written, well structured, and correctly illustrated with figures.

I congratulate the authors on their good work and encourage them to incorporate a few suggestions for improvement.

1. Present the modeling domain. It would be interesting for the reader to know some basic data of the modeling domain: extension of the city of Braunschweig, number of inhabitants,

population density, orography, climate, identified flood risks, drainage system, flood protection, type of economic activities that are developed, etc.

**REPLY:** Thank you for the suggestion. We have now added a separate section (2.1) to provide some description on the study domain.

2. Describe the time scale of the flood being analyzed. If the phenomenon is primarily caused by pluvial flooding, the time available between the onset of precipitation and the occurrence of damage is likely to be relatively short. While the analyses presented seem to focus on the moment of maximum flood extent, an early warning system requires understanding how far in advance potential damage can be predicted. It would be valuable if the authors could provide insights or comments on this aspect.

**REPLY:** Thank you for this comment. While our analysis primarily focuses on the maximum flood extent and water depth, we recognize the importance of considering the time scale from precipitation onset to the occurrence of flooding in the context of early warning systems. We have now extended our discussion to also include the lead times of different precipitation products and the practical integration of RIM2D into operational early warning systems. With regards to the time when the damage occurs, calculating exact times is a little bit challenging. As discussed in the Impact Calculation section, damage calculation is not just dependent on water depth in combination with velocity, but also other factors such as inundation duration, preparedness, early warning, etc. Further on the time of damage will not be uniform in the whole domain and would vary in different locations. This is why the study is focused on the hazard, i.e.inundation only. But this is in the end an indicator when and where damages occur.

3. Comment on the early warning system into which this tool could be integrated. While simulation accuracy and computation times are crucial, they are not the only requirements for incorporating such tools into early warning systems. To evaluate the utility of this tool within such a system, it is essential to thoroughly characterize the system by addressing several key aspects:

Define the system's objectives (monitoring, decision support, emergency warning, ...).

Describe the frequency and cadence of radar data updates. The text specifies that radar data are available every hour. How long does it take to have rainfall data for the previous hour available?

Specify whether any nowcasting methods are in place to predict precipitation in advance. This may gain some lead time for the warning.

Outline the proposed real-time workflow, including data processing and integration. What is the actual time window for the 2D flood inundation model to allow for a timely warning? Account for the time required to generate results, understand them, and disseminate critical information.

In summary, assess whether the system as a whole can meet its mission goals within the required lead-times to effectively protect the population. A discussion of these factors would provide valuable insight into the practical integration and effectiveness of the tool in operational early warning systems.

**REPLY:** We value the reviewer's insightful comments on the requirements for integrating RIM2D into an early warning system. In response, we have expanded our discussion to include an overview of RIM2D's integration into such a system, covering its objectives, radar data frequency, the potential use of nowcasting techniques, and the real-time workflow.

**Specific suggestions**

Line 84-86: The authors indicate that buildings are supposed to act as reflective boundaries. What about other types of obstacles present in the city, such as street furniture, parked or circulating cars? In some floods these elements may cause important obstructions to the flow. Is that considered?

**REPLY:** As mentioned, buildings within the simulation domain are treated as reflective boundaries in the model (to be precise: all obstacles represented in the DEM, i..e permanent obstacles), while other urban elements, such as street furniture and parked vehicles, which may influence flow dynamics, are not explicitly incorporated. This exclusion is mainly due to the lack of data and the challenges in obtaining such information for a flood event, particularly when it comes to forecasts. There is no such thing as an operational monitoring of parked cars or other non-permanent flow obstacles. We have included statements in the manuscript on this problem, which is in fact system-immanent.

Line 253. I am not sure this information on running the model with multiple GPUs fits in the conclusions section, since it was not covered in the results or discussion sections. I would suggest presenting it earlier.

**REPLY:** Thanks for the suggestion. This information is now moved to the Computational performance and runtime section.

Typos:

From the formal standpoint, the paper is correctly organized and well written. However, I noticed a few typos:

*Line 55 With-> with*

*Line 57 furthermore->Furthermore*

Line 99 RADOALN->RADOLAN

*Line 110 provides->provide*

Line 192 were->where

**REPLY:** Thank you for highlighting these issues. All the identified mistakes have now been corrected.

---

## Author Response (AR2)

Dear Prof. Maria Carmen Llasat,

Thank you for your detailed and constructive feedback regarding our manuscript.

We have now addressed the first two points you highlighted:

- We have ensured consistency in the use of "RIM2D" throughout the text.
- We have clarified that the RADOLAN data refers to the accumulated precipitation over the previous hour, as estimated from radar observations.

After careful consideration, we believe the content and structure of our work are more aligned with the format of a regular paper. Therefore, we would prefer to proceed with the publication as a regular paper.

We appreciate your time and guidance throughout this process and look forward to your response.

Best regards,
Shahin Khosh Bin Ghomash and coauthors